# The antidepressant sertraline provides a novel host directed therapy module for augmenting TB therapy

Deepthi Shankaran[1,2†], Anjali Singh[1,2†], Stanzin Dawa[1,2], Prabhakar Arumugam[1,2], Sheetal Gandotra[1,2], Vivek Rao[1,2*]

[1]CSIR- Institute of genomics and Integrative Biology, New Delhi, India; [2]Academy of Scientific and Innovative Research, Ghaziabad, India

**Abstract** A prolonged therapy, primarily responsible for development of drug resistance by *Mycobacterium tuberculosis* (Mtb), obligates any new TB regimen to not only reduce treatment duration but also escape pathogen resistance mechanisms. With the aim of harnessing the host response in providing support to existing regimens, we used sertraline (SRT) to stunt the pro-pathogenic type I IFN response of macrophages to infection. While SRT alone could only arrest bacterial growth, it effectively escalated the bactericidal activities of Isoniazid (H) and Rifampicin (R) in macrophages. This strengthening of antibiotic potencies by SRT was more evident in conditions of ineffective control by these frontline TB drug, against tolerant strains or dormant Mtb. SRT, could significantly combine with standard TB drugs to enhance early pathogen clearance from tissues of mice infected with either drug sensitive/tolerant strains of Mtb. Further, we demonstrate an enhanced protection in acute TB infection of the highly susceptible C3HeB/FeJ mice with the combination therapy signifying the use of SRT as a potent adjunct to standard TB therapeutic regimens against bacterial populations of diverse physiology. This study advocates a novel host directed adjunct therapy regimen for TB with a clinically approved antidepressant to achieve quicker and greater control of infection.

**\*For correspondence:**
vivek.rao@igib.in

[†]These authors contributed equally to this work

**Competing interest:** The authors declare that no competing interests exist.

## Editor's evaluation

Host-directed therapies have the potential to improve the management of tuberculosis by shortening the duration of chemotherapy and promoting recovery of respiratory sufficiency. In this useful study, the authors investigate the utility of sertraline as a potential host-directed therapy. They provide solid evidence that sertraline potentiates the activity of anti-tubercular drugs in macrophages as well as in the murine model of tuberculosis infection. The study will be of interest to tuberculosis researchers and infectious disease specialists.

## Introduction

The current TB therapy regimen ranging between 6 months for pulmonary and 1–2 years for extra pulmonary infections, is often associated with severe drug-induced toxicity in patients. Moreover, its failure to completely eradicate the pathogen from the host, forms an ideal platform for the emergence of drug-resistant strains (*Blumberg et al., 2003*; *Sharma et al., 2017*). It is not surprising that these strains have emerged at an alarming rate in the population and are imposing serious impediments to TB control programs globally (*Falzon et al., 2017*; *Singh et al., 2020*). Introduction of newer modalities like host directed therapy (HDT) with the potential to reduce duration of therapy and not be affected by pathogen resistance mechanisms offer significant advantages in this scenario (*Hancock*

*et al., 2012*; *Munguia and Nizet, 2017*). Effective molecular entities like antibodies, cytokines, cell-based therapies, repurposed drugs have been tested against bacterial and viral infections (*Parida et al., 2015*; *Li et al., 2019*; *Foster, 2010*; *Zakaria et al., 2018*; *Fedson et al., 2015*; *Skerry et al., 2015*; *Scanlon et al., 2015*; *Yedery and Jerse, 2015*; *Jiménez de Oya et al., 2018*). Several facets of infection response ranging from enhancing pathogen clearance to augmenting host metabolism or nutrition have been tapped to develop novel host targeted interventions strategies against complex bacterial infections (*Lange et al., 2019*; *Phelan et al., 2018*; *Oh et al., 2006*; *Cusumano et al., 2011*; *Berube and Bubeck Wardenburg, 2013*; *Blum et al., 2015*).

Mtb infection invokes several mechanisms of pathogen clearance in host cells involving, the induction of pro-inflammatory response, metabolic stress, phago-lysosomal lysis programs, apoptosis/autophagic mechanisms (*Benmerzoug et al., 2018*; *de Martino et al., 2019*; *Shi et al., 2019*). On the other hand, by virtue of its long standing association with humans, Mtb has evolved complex and intricate mechanisms to survive and establish optimal infection in the host (*Liu et al., 2017*; *Padhi et al., 2019*; *Dey et al., 2017*; *Gupta et al., 2017*; *Brites and Gagneux, 2015*; *Lin et al., 2016*; *Smith et al., 2019*; *Mehta and Singh, 2019*). While attempts to boost the host immune mechanisms for better control of the pathogen are promising, efforts have focused on the development of counter-measures against pathogen mediated subversion of cellular clearance mechanisms (*Salahuddin et al., 2013*; *Suárez-Méndez et al., 2004*; *Singhal et al., 2014*; *Naftalin et al., 2018*; *Lachmandas et al., 2019*; *Rayasam and Balganesh, 2015*).

We hypothesized that neutralizing a prominent pathogen-beneficial response would indirectly supplement host immunity facilitating better pathogen control. The early, robust, type I IFN response of phagocytes to intracellular bacterial infections including Mtb, is often associated with a detrimental effect on host immune activation and survival (*Dorhoi et al., 2014*; *Watson et al., 2015*; *Donovan et al., 2017*; *Shankaran et al., 2019*). We sought to offset this response by using sertraline (SRT) – a previously identified antagonist of poly I:C mediated type I IFN signaling, in macrophages and evaluate Mtb infection dynamics (*Zhu et al., 2010*). We demonstrate that SRT, effectively inhibited infection induced IFN that manifested as growth arrest of Mtb in macrophages. Interestingly, SRT could augment mycobacterial killing in the presence of INH (H) and rifampicin (R), two of the frontline TB drugs in macrophages by effectively lowering the concentration of antibiotics required to achieve clearance. Remarkably, the combination proved effective even against dormant bacilli or antibiotic tolerant Mtb strains. Addition of SRT to TB drugs – HR or HRZE (HR +pyrazinamide, ethambutol) significantly protected infected mice from TB-related pathology both by enhancing bacterial clearance and host survival, implying on the usefulness of this combination therapy in both the intensive and continuation phases of anti TB therapy (ATT). Taken together, we report a novel adjunct TB therapy module by repurposing the FDA-approved antidepressant – sertraline.

## Results

### Sertraline augments control of Mtb by frontline TB drugs in the macrophage model of infection

Macrophages respond to Mtb infection by elaborating an array of signaling cascades and effector functions with the nucleic acid driven type I IFN response as an active and dominant response to during infection in macrophages (*Shankaran et al., 2019*; *Liu et al., 2017*; *Xu et al., 2014*; *Petit-Jentreau et al., 2017*; *Wassermann et al., 2015*; *Wiens and Ernst, 2016*; *Moreira-Teixeira et al., 2018*). With its proven benefit to the pathogen, we hypothesized that suppressing the type I IFN response in cells would alter macrophage infection dynamics. The FDA-approved antidepressant – SRT, previously identified to suppress TLR-mediated IRF signaling in macrophages, controlled Mtb-induced type I IFN response in a dose-dependent manner (*Figure 1A*). While extremely low doses up to 0.5 µM did not alter the response, doses of 1 µM and higher were significantly inhibitory. Even at 1 µM, SRT inhibited the response by 1.5–2 folds that increased to 35 and 43%, at 5 µM and 10 µM of SRT, respectively (*Figure 1A*). At 20 µM, the observed fivefold decrease in RLU SRT reflected as complete loss of IFNβ secretion by Mtb infected macrophages (*Figure 1B*). Moreover, while naive macrophages were permissive for Mtb growth to 10-folds of input by day 6, 20 µM of SRT, despite its minimal activity on Mtb in vitro (inhibiting Mtb growth by ~30–40% by days 8–11 of culture), efficiently restricted growth in infected macrophages (*Figure 1C*).

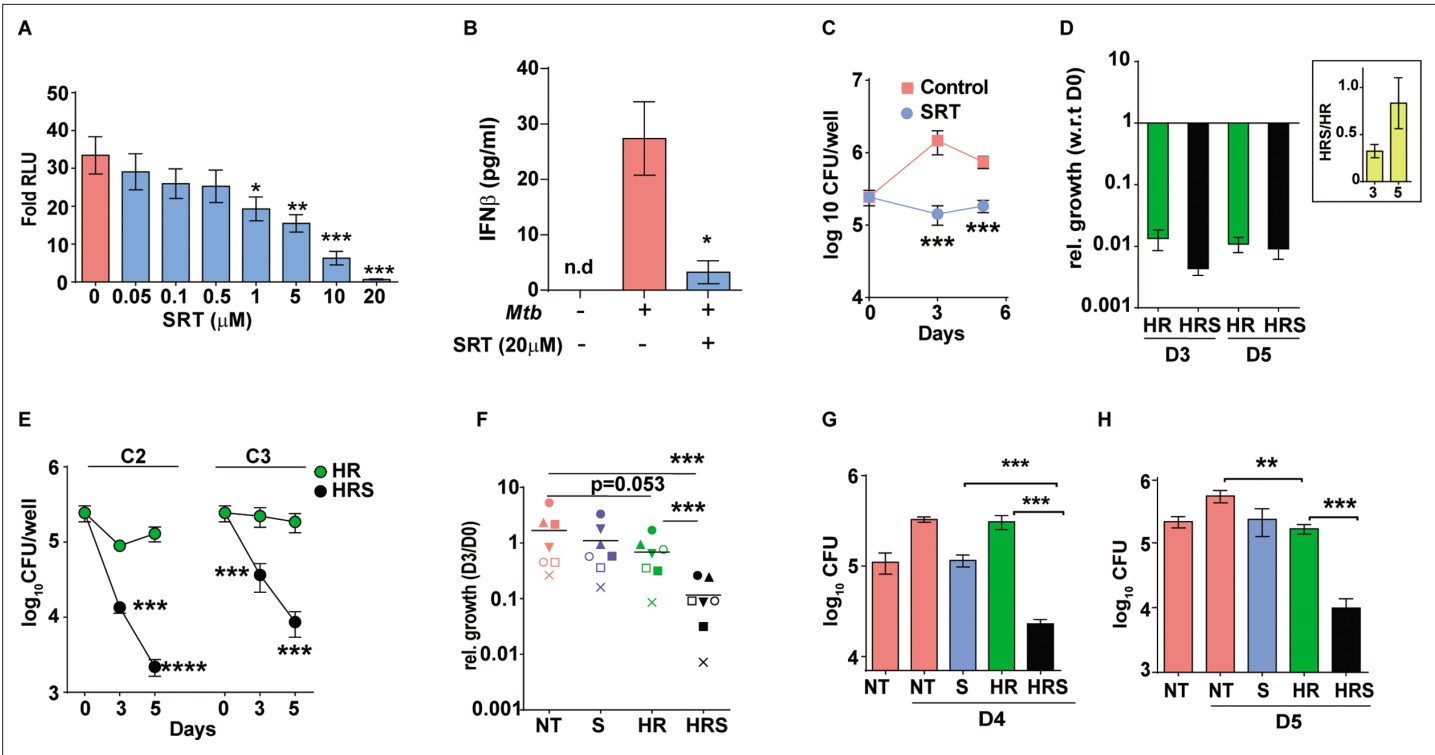

**Figure 1.** Sertraline inhibits Mtb-induced Type I IFN response and restricts intra-macrophage Mtb growth. (**A**) IRF-dependent luciferase activity in THP1 Dual macrophages following infection with Mtb at a MOI of 5. Cells were left untreated or treated with increasing concentrations of SRT for 24 hr in culture and the luminescence in culture supernatants was measured and is represented as mean ± SEM from three independent experiments with triplicate wells each. (**B**) Levels of IFNβ in cell supernatants of Mtb infected or naïve THP1 macrophages after 24 hr of infection. Cells were left untreated or treated with SRT for 24 hr and cytokine levels were measured by ELISA and is represented as mean ± SEM from three independent experiments with triplicate wells each. (**C–H**) Intracellular bacterial numbers in THP1 Dual macrophages following infection with Mtb at MOI5 for 6 hr and then either left untreated (NT) or treated with, Sertraline (SRT/ S), HR or a combination of all three (HRS), data represents mean± SEM from N=3 replicate experiments. (**C**) counts (CFU) at day 3 and day 5 post infection in untreated or SRT-treated macrophages, (**D**) Growth in cells treated with HR at 200ng/ml INH and 1000ng/ml Rif (C1). The relative bacterial counts at day 3 and day 5 post infection with respect to day 0 (6hp.i.) is represented. (**E**) Growth of Mtb in cells treated with 20ng/ml INH and 100ng/ml Rif [C2] or with 8ng/ml INH and 40ng/ml Rif [C3] is represented as mean CFU± SEM of N=3 independent experiments. (**F**) Mtb growth in primary human M1- differentiated MDMs from PBMC of seven individuals is represented as CFU relative to day 0. Macrophages were infected at a MOI of 5 for 6 hr and treated with SRT, HR, HRS or left untreated. Each symbol represents one individual, the relative growth at day 3 with respect to day 0 is depicted. (**G–H**) Growth in THP1 macrophages treated with Vit. (**C**) for 24 hr post infection (**G**) or with 200 µM Oleic acid for 48 hr prior to infection (**H**) and treatment with HR at C2 concentration with and without SRT. Except for E, paired t-test comparing ratios, other datasets were compared with unpaired t-test; **p<0.01, ***p<0.001.

This observed bacterial stasis in SRT-treated macrophages prompted us to analyze the effect of this treatment in conjunction with standard TB drugs. Decreased portioning of potent antibiotics to regions of bacterial presence in the center of granulomas remains a major cause for the improper clearance of bacteria from infected tissues (*Cicchese et al., 2020*; *Prideaux et al., 2015*). In addition, Mtb is endowed with the inherent propensity to enter into a drug-tolerant, non-replicating dormant state associated with drug tolerance and antibiotic failure (*Sarathy et al., 2018*). In order to mimic conditions of reduced effective drug concentrations, we tested the effect of SRT across three different concentrations of frontline TB drugs – isoniazid (H) and rifampicin (R). As expected, a gradual reduction in antibiotic effectivity was observed with decreasing doses; while C1 (200 ng/ml INH and 1000 ng/ml Rifampicin) was able to reduce bacterial numbers drastically (~62-fold by day 3 and ~92-fold by day 5), a 10-fold lower dose of C2 (20 ng/ml INH and 100 ng/ml Rifampicin) was less effective resulting in with 2–3 folds lower numbers and bacteriostasis in cells treated with the 2.5-fold lower dose -C3-8 ng/ml INH and 40 ng/ml Rifampicin (similar to input values) (*Figure 1D and E*). Addition of SRT efficiently boosted the bactericidal properties of antibiotics at all concentrations tested. Even at the highly effective HR concentration, addition of SRT further reduced bacterial numbers by 2–3 folds (*Figure 1D*). Surprisingly, the ability of SRT to boost antibiotic properties was evident at lower doses- at the C2

dose, SRT lowered numbers by 8 at day 3 and 50 folds by day 5. Even at the lowest dose of C3, SRT substantially enhanced bacterial killing by 6- and 20-folds at days 3 and 5 post infection, respectively, in comparison to HR (*Figure 1E*).

This ability of SRT was also observed in primary macrophages (human monocyte derived macrophages) from seven healthy individuals. Although Mtb growth varied across samples, the antibiotic potentiating effect of SRT was preserved (*Figure 1F*). Across different individuals, while SRT and HR individually showed minimal but highly variable bacterial control, SRT universally synergized with antibiotics to further reduce bacterial numbers by 10–15 folds supporting a more general adjunct activity of SRT to frontline TB drugs.

For further test of the combination in conditions that reflect reduced antibiotic availability, we used two recently established models of Mtb infection that would mirror this in vivo situation– (1) the recently reported model of vitamin-C-treated THP1 macrophages associated with Mtb dormancy and loss of HR efficacy (*Sikri et al., 2018*) and (2) lipid-rich conditions that restrict entry of frontline TB drugs in vivo (*Dartois, 2014*; *Jaisinghani et al., 2018*). Again, while HR efficacy was completely nullified in macrophages with vitamin C pretreatment, inclusion of SRT resulted in 10–12-fold lower bacterial numbers after 4 days of treatment (*Figure 1G*). In order to mimic lipid rich conditions in ex vivo studies, we treated THP1 macrophages with 200 μM oleic acid and observed significant lipid loading in cells without any alteration in cellular morphology. However, the combination of H and R showed poor efficacy in controlling infection, from and found poor efficiency of HR in these cells, addition of

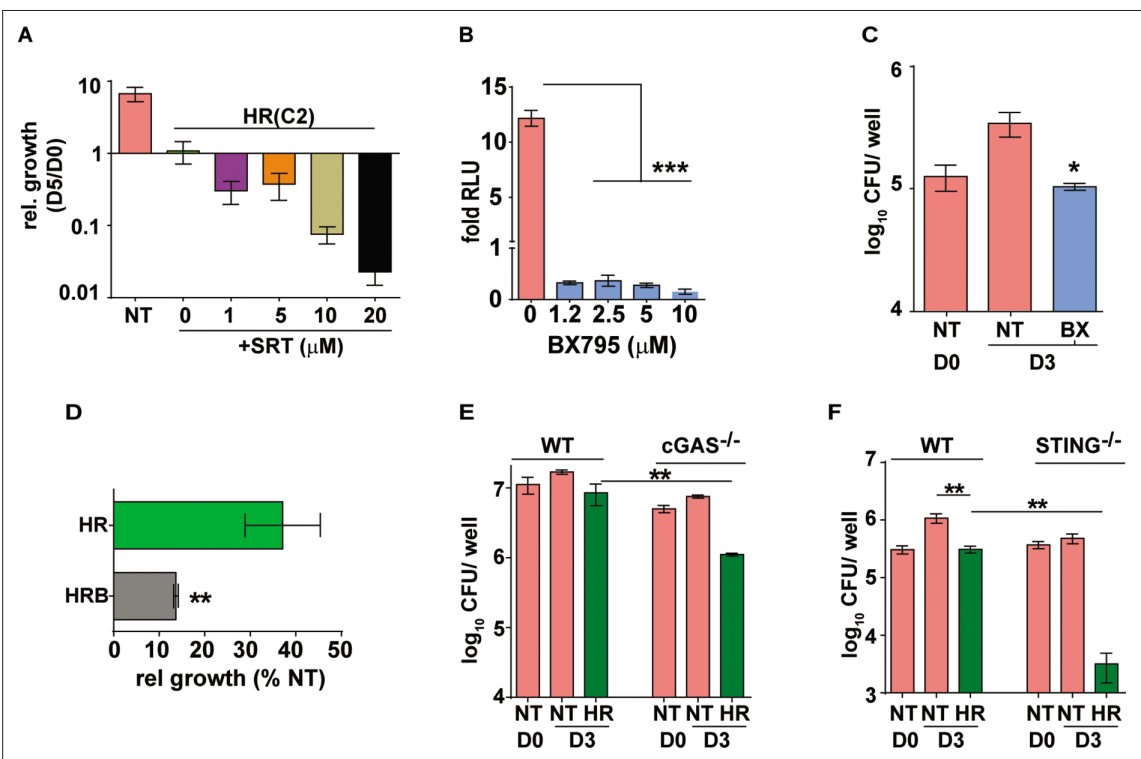

**Figure 2.** Augmentation property of SRT is due to its ability to inhibit IFN signaling. (**A**) Bacterial growth in macrophages treated with HR at C2 and different concentrations of SRT as indicated. Values represented are mean relative CFU at day 5w.r.t day 0 ± SEM of triplicate assays from N=3 independent experiments (**B**) IRF-dependent luciferase activity in THP1 dual macrophages 24 hr after treatment with varying doses of BX795 along with infection with Mtb at MOI of 5. (**C, D**) Bacterial growth in macrophages left untreated or treated with 10μM BX795. Relative growth of Mtb in macrophages treated with HR and HR+BX795 (HRB) for 3 days (**C**), The percentage relative growth of intracellular bacterial numbers in HR or HRB groups with respect to untreated samples is depicted (**D**). (**E, F**) Bacterial growth in murine RAW 264.7 derived macrophages ISG (WT), cGAS[-/-], STING[-/-] that were left untreated or treated with HR for 3 days is shown as mean CFU ± SEM for triplicate wells of N=2 (**E**) and mean ± SEM for (**F**) of N=2/3 experiments. Statistical significance by unpaired t-test- *p<0.05, ***p<0.001 is indicated.

The online version of this article includes the following figure supplement(s) for figure 2:

**Figure supplement 1.** Bacterial growth in macrophages treated with HR alone or in combination with low doses of SRT as indicated.

**Figure supplement 2.** Effect of sertraline on in vitro Mtb cultures with and without HR.

SRT led to more than 10-fold reduction in bacterial loads, further substantiating the increased bactericidal properties of the combination (HRS) in conditions of decreased antibiotic efficacy (*Figure 1H*).

## Sertraline-mediated increase in anti-microbial effect of TB drugs is dependent on attenuation attenuates Mtb-induced type I IFN signaling in macrophages

Similar to the type I IFN response restriction, SRT was found to increase HR efficiency in a dose dependent manner with doses lower than 1 μM failing to alter the growth of Mtb in macrophages (*Figure 2—figure supplement 1*). At concentration of 1–20 μM, SRT, could effectively synergize with antibiotics and inhibit bacterial numbers by ~5–70 folds, respectively, in the treated macrophages (*Figure 2A*). However, SRT along with HR does not significantly inhibit growth in vitro, even 6 μM SRT failed to impact Mtb growth either alone or in combination with 3 concentrations of HR (*Figure 2—figure supplement 2*), alluding to an indirect mode of SRT in enhancing the ability of macrophages to control bacteria rather than a direct action on bacteria. Our observation that SRT controlled Mtb induce type I IFN responses argued for further evaluation of this axis as a plausible mechanism. We first evaluated the impact of BX795, a specific inhibitor of type I IFN signaling, on bacterial growth control in our model of infection (*Figure 2B–D*). We first established the efficacy of BX795 in abrogating Mtb-induced type I IFN in THP1 macrophages at different concentrations. Treatment of macrophages with BX795 at concentrations as low as 1.2 μM completely nullified the infection induced type I IFN response in macrophages (*Figure 2B*). Again, similar to SRT, BX795 alone resulted in bacteriostasis with bacterial numbers maintained at input levels in these cells in similar to SRT treated macrophages (*Figure 2C*). Again, addition of BX795 to HR enhanced bacterial control by 4–5 folds by day 3 of infection as compared to the antibiotics alone (*Figure 2D*). Given our observation of increased antibiotic efficacy by IFN inhibition (by SRT and BX795), we reasoned that a similar effect of increased antibiotic efficacy would be visible in IFN signaling deficient macrophages. To test this, we compared the intracellular bacterial numbers in cGAS$^{-/-}$ and STING$^{-/-}$ macrophages with WT cells after 3 days of treatment with HR. As expected, Mtb was restricted in its growth in these macrophages. Nearly 2–3 folds lower bacterial numbers were observed in cGAS$^{-/-}$ (*Figure 2E*) with a much stronger inhibition (40–50 folds) in STING$^{-/-}$ macrophages (*Figure 2F*) in comparison to the corresponding IFN-sufficient cells.

## Activation of host cell inflammasome is critical for sertraline-mediated increase of antibiotic efficacy

Elaboration of a pro-inflammatory response is one of the potent responses with activation of anti-mycobactericidal activities in macrophages. Our results hinted at the reciprocal effect of IFN signaling on macrophage infection control as has been demonstrated earlier (*Teles et al., 2013*). Further support for this was observed in the macrophages preactivated with IFNγ and treated with HR or HRS. While IFNγ treatment resulted in two- to threefold lower bacteria in combination with HR alone (*Figure 3—figure supplement 1*), we found a similar enhancement in IFNγ mediated control in SRT-treated macrophages (*Figure 3A*). Analysis of the pro-inflammatory response in SRT-treated infected macrophages also lend credence to this hypothesis. Gene expression profiles showed reduction in the expression of TNF and IL1β after 18 hr of treatment with SRT consistent with reported literature (*Figure 3B–E*; *Sitges et al., 2014*). In comparison to HR, while TNF expression was reduced slightly in the HRS-treated macrophages (*Figure 3B*), negligible levels of TNF was observed in the cell supernatants in SRT-treated cells (SRT, HRS), contrasting with the high levels of TNF (between 500pg- 1 ng/ml) in the case of Mtb infected cells with or without treatment with HR (*Figure 3C*). Surprisingly, while expression levels of IL1β were 2–3 folds lower in HRS-treated macrophages by 18 hr (*Figure 3D*), the amount of secreted cytokine was significantly elevated (>2–3 folds) in macrophages treated with SRT/HRS from 18 hr until 66 hr (*Figure 3E*). These data strongly indicated potential activation of the host cell inflammasome by SRT. With several studies supporting inverse regulation of type I IFN and inflammasome activation (*Szałach et al., 2019*; *Ruiz-Grosso et al., 2020*), we tested the efficacy of SRT to potentiate antibiotic mediated killing in the presence of inflammasome inhibitors- isoliquiritigenin (I) and MCC950. Isoliquiritigenin did not show any significant effect on macrophages that were untreated or treated with HR or SRT alone (*Figure 3F*, inset). However, while addition of SRT to HR significantly reduced bacterial numbers in macrophages by more than 10–20 folds, pretreatment of cells with the isoliquiritigenin, completely nullified this boosting effect of SRT on antibiotic efficacy (*Figure 3G*).

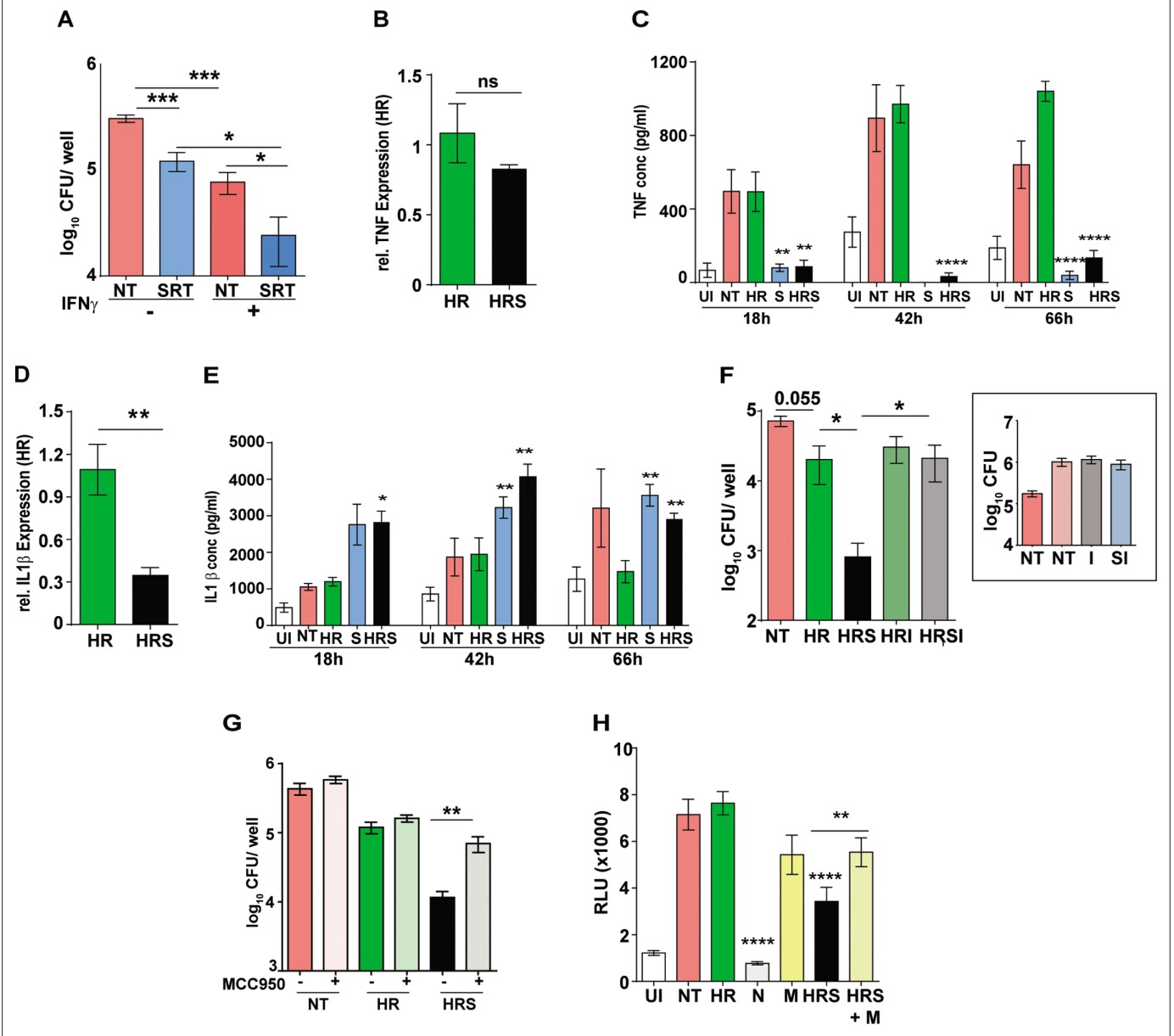

**Figure 3.** Important role for inflammasome activation in antibiotic potentiation by SRT. (**A**) Growth of Mtb in macrophages pre-treated with IFNγ for 16 hr, infected with Mtb for 6 hr and then left untreated or treated with SRT for 3 days. Mean CFU values for triplicate assay wells from two independent experiments (N=2) ± SEM is shown. The relative expression of inflammatory cytokines in macrophages treated with HR or HRS. Data in B and D depict fold expression (transcript abundance) relative to HR alone as mean ± SEM from two independent experiments with duplicate wells each at 18 hr post treatment. (**C**) and (**E**) depict secreted cytokines at indicated time points post treatment. (**C**) Average values ± SEM of TNF at 18, 42, and 66 hr post treatment of two independent experiments of triplicate wells (N=2). (**E**) IL1 β levels in cell supernatants at 18, 42, and 66 hr of triplicate wells (n=3). (**F, G**) Growth of Mtb in macrophages in the presence of inflammasome inhibitors. Macrophages were infected with Mtb for 6 hr and then left untreated (NT) or treated with HR and HRS with and without - isoliquiritigenin-I (**F**) or MCC950 (**G**) for 5 days. Bacterial growth in macrophages treated with I alone is shown in the inset. Mean CFU / well ± SEM of N=3 assays in triplicate wells are depicted. (**H**) IRF-dependent luciferase activity at 24 hr in Mtb infected THP1 dual macrophages (MOI –5) and treated with Nigericin (N) to activate or MCC950 (M) to inhibit inflammasomes. Mean RLU ± SEM of triplicate assays of three independent experiments is shown. Data is represented as mean ± SEM for two to three independent experiments containing triplicate wells per assay. Statistical significance by unpaired t-test- *p<0.05, ***p<0.001 is indicated.

The online version of this article includes the following figure supplement(s) for figure 3:

**Figure supplement 1.** Bacterial growth in naive macrophages or pre-treated with IFNγ.

Treatment with a NLRP3-specific inhibitor- MCC950 also resulted in a reversal of enhanced bacterial control in HRS in infected macrophages without altering the effect of antibiotics alone on the bactericidal properties validating the involvement of inflammasome in the ability of SRT to enhance bacterial control by frontline antibiotics. In line with previous studies (*Labzin et al., 2016*; *Yu et al., 2018*), we also observed a significant decrease in Mtb-induced type I IFN in macrophages following treatment with inflammasome activation (*Figure 3H*). While treatment with SRT and HR significantly reduced this response in macrophages, pretreatment with the inflammasome inhibitor- MCC950 partially reversed this decline again arguing for an important role for inflammasome activation in the type I IFN modulating effect of SRT.

## Sertraline augments frontline drugs in acute model of Mtb infection

The antibiotic escalating properties of SRT in macrophages prompted testing of this combination regimen in vivo. We reasoned that the survival of Mtb infected C3HeB/FeJ mice would provide an optimal platform for a fast readout of comparative drug efficacies of standard TB drugs and the adjunct regimen with SRT. Given the inherent heterogeneity of infection dynamics across animals of this mouse strain (*Dawa et al., 2021*), we opted for the higher dose of Mtb Erdman infection (500 cfu/ animal) to achieve acute infection in all animals. The adjunct effect of SRT was tested at effective-C1(1 x) and ineffective doses - C2 (0.1 x) and C3 (0.01 x) of INH and Rifampicin ad libitum in drinking water. To facilitate disease progression prior to treatment initiation, treatment with antibiotics was initiated after 2 weeks of infection (*Figure 4A*). Aerosol delivery at this dose resulted in precipitous disease with rapid killing of untreated (NT) animals by day 31 of infection (*Figure 4B*). SRT alone was effective in delaying the disease progression in animals as the mean survival time (MST) increased from 31 days for untreated animals to 38 days. HR at the lowest concentration (C3), similar to SRT, deferred animal mortality with MST of 41 days while the higher doses of HR (C2 and C1) increased MST significantly to 85 and >90, respectively. Given the higher susceptibilities of female mice to infection observed in our study, we decided to differentially tally the gender-based effects of SRT treatment in these animals. The combination of HRC3 and SRT, nearly doubled the MST of both female and male mice to 78 days and 100 days from 41 and 45 days respectively (*Figure 4C*). This benefit of SRT was similar to that observed for a 10-fold higher concentration of HR alone (MST of 85 for HRC2). The advantage of SRT inclusion with HR was evident in the gross lung pathology by 30 days of infection. Both NT and HRC3 treated lungs showed extensive progressive granulomas, contrasting with a significant amelioration of pathology seen in HRC3 +S treated animals (scale bars of 0.1mm) (*Figure 4D*).

Further, we wanted to explore temporal benefits of the combination regimen after withdrawal of a limited-term treatment (*Figure 4E*). In this model, 100% of animals survived after treatment with HR(C1) at 16 weeks post infection (*Figure 4F*). A 7 week ad libitum treatment with the 10-fold lower dose of HR (HRC2) (*Figure 4F*) significantly decreased the MST of animals to 85 days with 100% mortality by 16 weeks. In contrast, 40% of HRC2S-treated animals, survived with a MST of 112 days for the group (*Figure 4F and G*). All animals with the highest dose of HR (HRC1) either alone or with SRT survived the infection. Despite, the significant heterogeneity of treatment response in male and female mice, with a relatively lower response as evidenced by the greater number of lesions in lungs of male animals, a co-operative effect of SRT inclusion was evident as a significant improvement in TB associated lung pathology (*Figure 4H*, *Figure 4—figure supplement 1*). Small macroscopic lesions were observed in lungs of 60% of the female mice treated with HRC1 that showed as multiple, well-defined granuloma in the H&E-stained sections by the 16[th] week post infection. In contrast, mice treated with SRT and HRC1 showed negligible involvement of the lung tissue in granulomatous cellular accumulation. Even in tissues of male mice, animals receiving the adjunct therapy showed fewer macroscopic and significantly lower numbers of microscopic granuloma in lung sections in comparison to animals treated with the antibiotics alone.

## Addition of sertraline increases bacterial control by frontline TB drugs

TB treatment in the intensive phase involves the use of 4 frontline TB drugs- HRZE for a period of 2 months and HR for an additional 4 months. Further to test the efficacy of SRT in combination with HRZE in an acute model of disease, Mtb infected C3HeB/FeJ mice were treated either with the established dose of HRZE or in combination with SRT (*Figure 5A*) and evaluated TB associated pathology of the lungs at 16 weeks of infection. Male (*Figure 5B*) and female (*Figure 5C*) animals treated with the 4

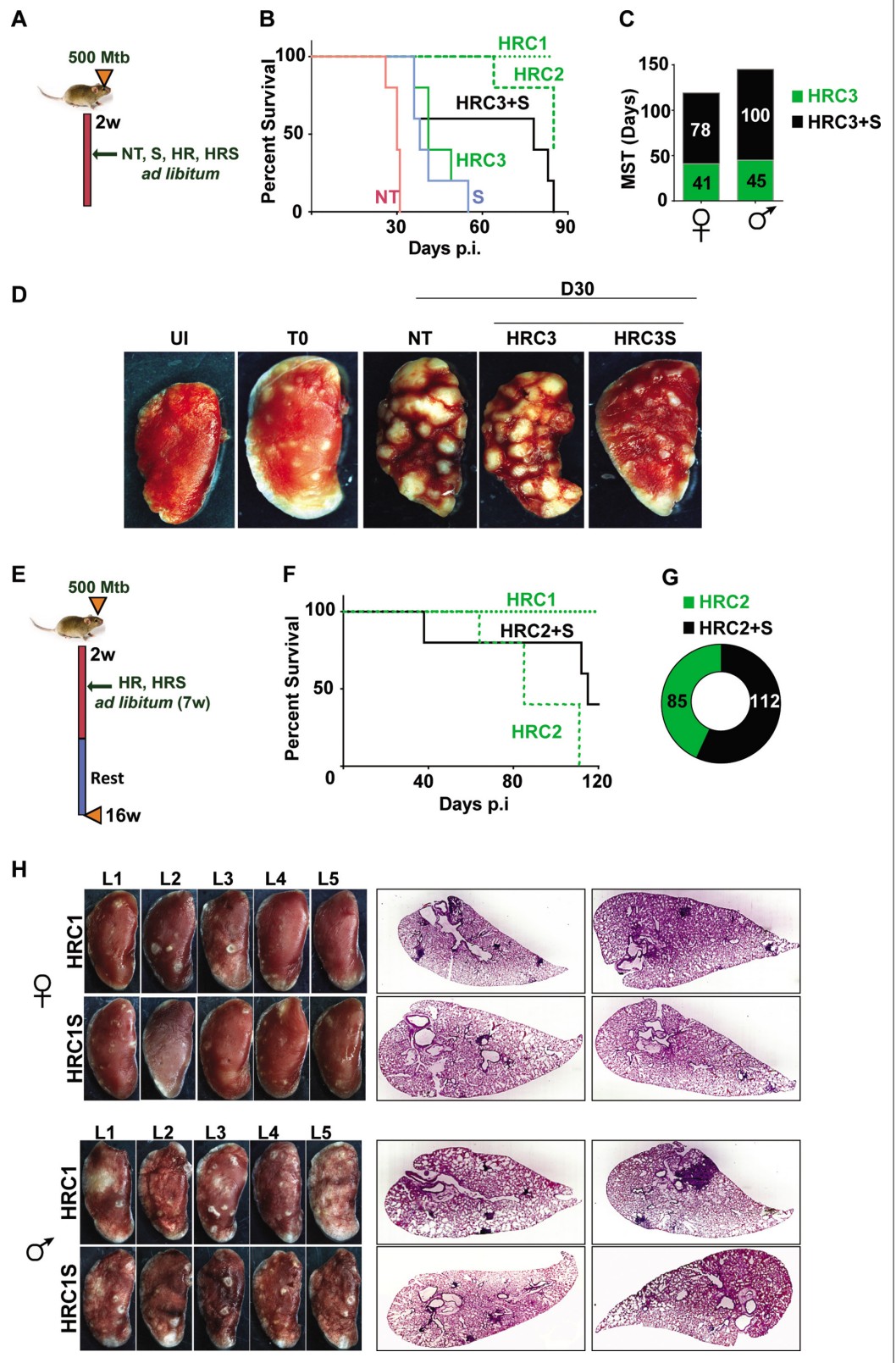

**Figure 4.** Adjunct SRT improves host survival in a susceptible mouse model of infection. (**A**) Schematic of Mtb infection and drug treatment in C3HeB/FeJ mice. (**B**) Survival curves of Mtb infected C3HeB/FeJ mice (5) left untreated (red) or treated with SRT alone (blue) or with different concentrations of H and R (green lines or boxes-HRC1- 1x: H-100μg/ml, R-40μg/ml, HRC2-0.1x, HRC3-0.01x) alone or along with SRT (10μg/ml)- black lines /

*Figure 4 continued on next page*

*Figure 4 continued*

boxes. (**C**) Median survival time of different treatment groups of mice (5 each of males and females). (**D**) Gross tissue morphology of lungs of uninfected animals (UI) and indicated groups at 30 days post infection with Mtb. (**E**) Schematic of infection and antibiotic treatment in C3HeB/FeJ with HRC2 and HRC1. (**F–G**) Survival (**F**) and MST (**G**) of C3HeB/FeJ mice treated with HRC2 or HRC2S. (**H**) Gross lung morphology at the end of 16 weeks and histological sections of lungs with H&E staining of C3HeB/FeJ mice either treated with HRC1 or in combination with SRT.

The online version of this article includes the following figure supplement(s) for figure 4:

**Figure supplement 1.** Quantitation of macroscopic lesions in left caudal lobe of lungs of animals infected with Mtb and treated with HR, HRZE or in combination with SRT.

drugs for 7 weeks harbored $10^5$–$10^6$ bacteria in their lungs, respectively. Addition of SRT appreciably improved drug efficacy, reducing the lung bacterial loads by a further 5–7 folds (*Figure 5B and C*). The drugs efficiently lowered tissue pathology as evidenced by the macroscopic lesions seen in the lungs of infected mice (*Figure 5D*). While both female and male mice showed small macroscopic lesions in lungs on treatment with HRZE, despite the heterogeneity between the genders, the combination of SRT and HRZE sufficiently decreased the extent of tissue involvement in TB associated pathology (Fig.

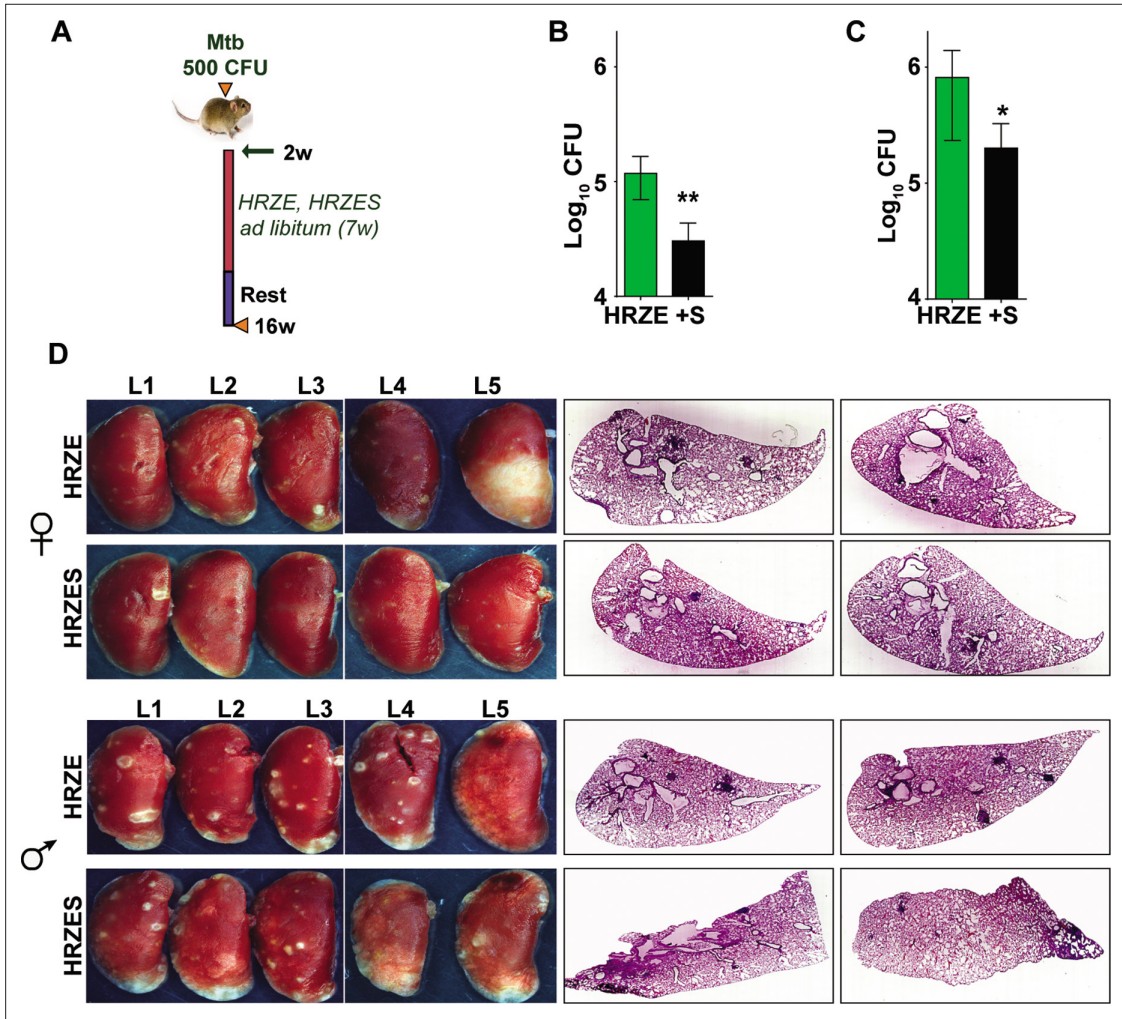

**Figure 5.** In vivo potentiation of SRT-mediated antimycobacterial activity. (**A**) Schematic of infected C3HeB/FeJ mice (10) and treated with HRZE HRC1- 1x: H-100µg/ml, R-40µg/ml, Z-150µg/ml, E-100µg/ml or in combination with SRT (10µg/ml). Animals were euthanized at the end of 16 weeks and extent of infection was determined by estimating bacterial numbers (CFU) in lungs of female (**B**) and male (**C**) mice. (**D**) Gross lung morphology and histochemical sections of lungs with H&E staining of C3HeB/FeJ mice infected with Mtb and treated either with HRZE or in combination with SRT. Statistical significance by unpaired t-test-*p<0.05, ***p<0.001 is indicated.

S1D). This difference was more evident in tissue sections, wherein animals treated with HRZES were devoid of granulomatous infiltrates in contrast to the HRZE treated animals which had significantly higher numbers of granulomas in the lungs (Scale bars of 0.1mm)(*Figure 5D*).

## Sertraline aids in faster clearance of infection in vivo

To evaluate the adjunct regimen for early bacterial clearance rates, we infected C57BL/6 mice with 500 CFU of Mtb and enumerated bacterial burdens upto 8 weeks post treatment according to the schedule shown in *Figure 6A*. Bacterial numbers in the lungs reached ~$10^7$ CFU by 4 weeks of infection (day 0 of treatment) and remained steady over the 6-week period in untreated animals. While treatment with HRZE was efficient in steadily reducing these numbers by ~100 folds, addition of SRT to the regimen significantly enhanced control by a further 2–3 folds. Moreover, the adjunct regimen was efficient in controlling the infection in spleens of infected mice (*Figure 6B*). Although HRZE reduced splenic bacterial numbers significantly by 6–7 folds, HRZES was more potent reducing bacterial numbers further by ~20 folds (60–70% vs ~2–5%) by 21 days of treatment (*Figure 6B*, inset). A similar degree of enhanced bacterial control (4–6 folds lesser bacteria) was observed in lungs and spleens of Balb/c mice treated with SRT as an adjunct to conventional 4 drug-therapy (*Figure 6—figure supplement 1*). Untreated animal lungs showed a gradual consolidation of the tissue with increasing amounts of granulomatous cellular infiltration by 6 weeks of infection. Treatment with HRZE was efficient in reducing this infiltration significantly by the 6$^{th}$ week of infection with nearly 1/6$^{th}$ of the tissue showing signs of cellular infiltration (*Figure 6C*). Lungs of mice receiving the combination showed better resolution of granulomas with significantly smaller regions of cellular collection by 3 weeks dispersed across the tissue, that was more or less absent from the tissues of mice by the 6th week of treatment. Treatment with HRZE for up to 8 weeks did not effectively clear the bacterial from the lungs of infected mice (scale bar of 0.1mm). While untreated animals harbored close to $10^6$ bacilli in the lungs, all treated mice had between 40 and 640 bacteria in the lungs (*Figure 6D*). In contrast addition of SRT, resulted in the absence of countable bacteria in lungs of 4/5 mice with one animal having 40 bacteria, similar to the animal with the best resolution of infection by the HRZE treated animals. Even the spleens of animals receiving the combination treatment did not show any detectable bacteria as opposed to the frontline TB drugs alone substantiating the ability of SRT to impart early bacterial clearance from animal tissues in comparison to the standard 4 TB drugs (*Figure 6D*, inset). Further evidence of faster clearance was evident at 9 weeks post treatment and even at lower doses of SRT. While Mtb persisted in animals given HRZE showing countable colonies in both lungs and spleen (2 out of 3 animals), the combination of SRT was successful in eliminating bacteria from animal tissues at the doses of 3 mg/kg and 2 mg/kg (*Figure 6E*). Interestingly SRT doses as low as 0.125 mg/kg was sufficient to clear bacterial loads in spleen from animals (*Figure 6E*, inset), the bacterial numbers in the lungs were more or less similar to the numbers in the HRZE group. Even at doses of 0.5mgkg and 1mgkg, SRT was more effective in controlling bacteria in the spleens and lungs of mice as compared to the HRZE group, again validating that the combination imparted faster in vivo bacterial clearance.

## Addition of sertraline improves control of tolerant bacteria in vivo

We then tested the efficacy of the adjunct therapy against a drug tolerant Mtb strain in cellular and murine models of infection. Previously, we had demonstrated that N73- a clinical Mtb strain belonging to the L1 ancient lineage showed increasing tolerance to INH and Rifampicin as opposed to the modern L3 and L4 lineage strains by virtue of expressing the complete MmpL6 operon (*Arumugam et al., 2019*). In THP1 cells infected with Mtb, HR at C2 concentration supported stasis and failed to decrease of bacterial numbers (*Figure 7A*). The combination of HR with SRT, significantly controlled infection and reduced bacterial numbers by ~10–15 folds by 3 days and ~100 folds by day5. The pattern of significantly greater bacterial control was also observed in primary human macrophages; again, a combination of HR and SRT reduced intracellular bacterial numbers by 5–50 folds in comparison to the drugs alone (*Figure 7B*). With a strong indication of SRT's ability to boost the efficacy of frontline TB drugs against tolerant Mtb, we tested its in vivo activity in the acute model of C3HeB/FeJ mouse infection in combination with HR (*Figure 7C*). As expected, the drugs were not efficient in controlling infection induced lesions in the lungs reducing bacterial numbers by fivefolds (*Figure 7D*). However, lungs of HRS treated animals harbored ~5–8 folds lower bacterial numbers than HR treated animals. The effect of the combination was again better in controlling bacterial numbers in spleens

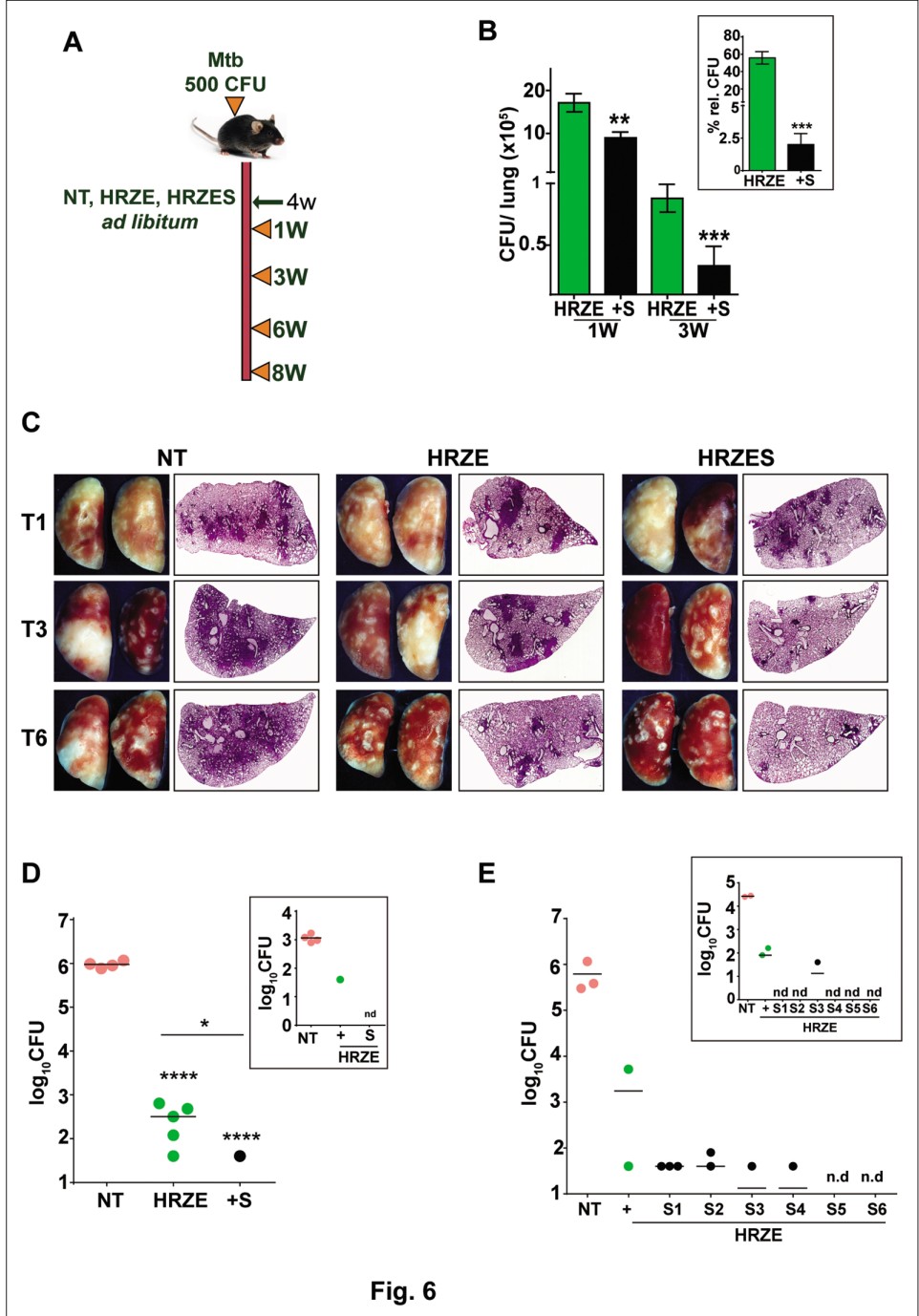

**Fig. 6**

**Figure 6.** SRT initiates early bacterial clearance in mice. (**A**) Schematic of Mtb infection and drug treatment in C57BL6 mice. Mice were infected with Mtb and treated with HRZE or HRZES (HRC1- 1X: H-100µg/ml, R-40µg/ml, Z-150µg/ml, E-100µg/ml, SRT 10µg/ml) treatment. Lung CFU post 1 and 3 weeks (**B**) and 3-week spleen CFU (inset) is represented as mean CFU ± SEM of N=3, (**C**) Gross lung morphology and H&E staining of tissue sections after treatment for the indicated number of weeks. (**D**) Lung and spleen CFU in Mtb infected C57BL6 mice (N=5), left untreated (NT) or treated with HRZE or HRZES for 8 weeks. (**E**) Lung and spleen CFU in Mtb infected C57BL6 mice (N=3) left untreated (NT) or treated with HRZE alone (+) or with HRZES (SRT at 6 concentrations (0.125mg/kg-S1, 0.25mg/kg-S2, 0.5mg/kg-S3, 1mg/kg-S4, 2mg/kg-S5 and 3mg/kg-S6)) for 9 weeks. Each individual dot represents an animal, nd- CFU not detected. Statistical significance by unpaired t-test-*p<0.05, ***p<0.001 is indicated.

The online version of this article includes the following figure supplement(s) for figure 6:

**Figure supplement 1.** Balb/c mice were infected with Mtb and the treated with HRZE or HRZES (HRC1- 1 X: H-100µg/ml, R-40µg/ml, Z-150µg/ml, E-100µg/ml, SRT 10 µg/ml).

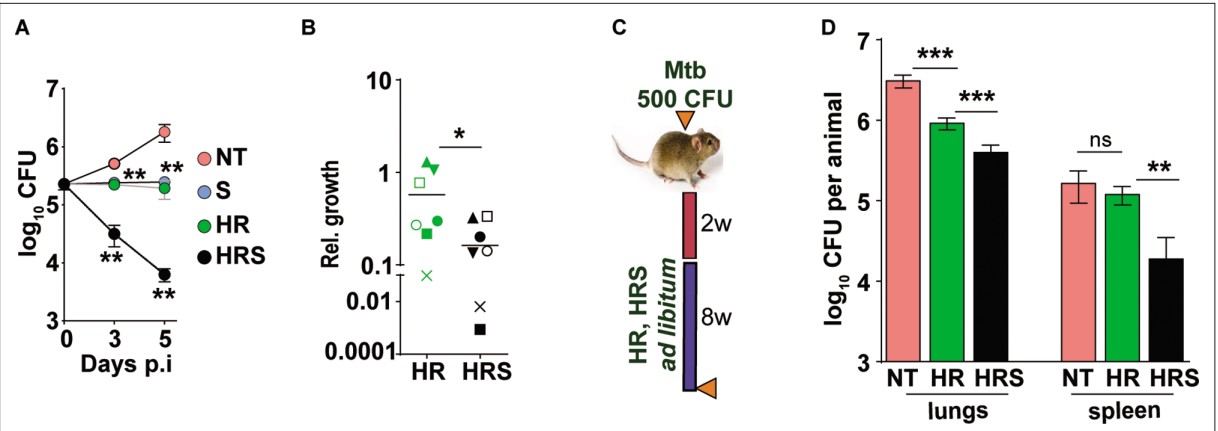

**Figure 7.** Addition of SRT helps better control of drug tolerant Mtb in vivo. (**A**) Intracellular bacterial growth in THP1 macrophages infected with HR tolerant Mtb strain at a MOI of 5 for 6 hr. Following this, cells were left untreated (NT) or treated with SRT, HR, or HRS for 3 or 5 days. Bacterial numbers were enumerated and is represented as average log10 CFU ± SEM from two independent experiments with triplicate wells each. (**B**) Bacterial numbers at day 3 of primary monocyte derived macrophages (M1) from seven independent donors. After 6h of infection, the macrophages were treated with HR and HRS for 3 days. The ratio of intracellular bacterial numbers in HR or HRS groups with respect to untreated samples is represented as relative growth with median values indicated by the horizontal line. (**C**) Schematic of Mtb infection and drug treatment in C3HeB/FeJ mice infected with Mtb for 2 weeks followed by treatment with 0.1X HR alone or with SRT (HRS) or 8 weeks. (**D**) Bacterial numbers (CFU) in lungs and spleen at the end of the experiment. Statistical significance by unpaired t-test -*p<0.05, ***p<0.001 is indicated.

of treated mice wherein HR did not reduce CFU in contrast to the six- to eightfold lower bacterial numbers with the combination.

## Discussion

Despite consistent efforts in identifying novel pathogen targeted interventions and streamlined pharmaceutical drug development control processes, fewer drugs have been accepted for clinical use in TB over the last 40 years (*Goel, 2014*). Repositioning existing drugs with established safety in humans is one of the quickest modes of developing effective control of infections that reduce the timeframe of regimen development. The need for an effective, short and pathogen-sterilizing regimen to tackle the growing problem of Mtb drug resistance and dormant bacterial populations has intensified efforts toward the development of host targeted therapies (*Palucci and Delogu, 2018*; *Mishra et al., 2019*; *Fatima et al., 2020*; *Dara et al., 2019*).

We and several other groups have identified type I IFN as an early response of host macrophages to infection with Mtb strains (*Donovan et al., 2017*; *Shankaran et al., 2019*; *Moreira-Teixeira et al., 2018*; *McNab et al., 2014*). With recent evidences implicating type I IFN as a pathogen beneficial response, we hypothesized that attenuating this axis would prove beneficial in controlling bacteria in macrophages. In line with this idea, we observed that the previously reported TLR3 antagonist – sertraline (SRT) could effectively stunt Mtb-induced type I IFN response in macrophages and also inhibit bacterial growth in macrophages.

SRT, along with other weak bases like fluoxetine, has been reported to moderately restrict intracellular Mtb replication in macrophages due in part to its weak basic nature without affecting the host cell viability (*Schump et al., 2017*). An important role for the acidic environment of bacteria resident vesicles (pH dependence) in the mycobactericidal properties of these drugs was demonstrated in this study. We also observed similar effects of SRT alone in our study – a moderate level of bacterial control in macrophages with minimal host cell death by treatment of SRT to infected macrophages. Additionally, several studies have indicated a direct action of SRT and other selective serotonin reuptake inhibitors on host immune response pathways: from enhancing the anti-inflammatory response (*Hannestad et al., 2011*), augmenting NK and CD8 cell response (*Benton et al., 2010*) and to inhibition of acid-sphingomyelinase (*Kouznetsova et al., 2014*), an essential component of the viral trafficking into NPC1 +endosomes in cells. Activation of host cell inflammasome and its antagonistic effect on the type I IFN response of cells has now been realized as important factors for controlling

bacterial infections (*Novikov et al., 2011*; *Ji et al., 2019*). A direct effect on activating eicosanoids which control type I IFN following infection was identified as a key bacterial clearance mechanism of infected cells (*Hawn et al., 2015*; *Mayer-Barber et al., 2014*). Consistent with this observation, targeted therapy toward elevating PGE2 activity protected mice from acute infection-induced fatality (*Mayer-Barber et al., 2014*). In line with these observations, we also describe the ability of SRT to potentiate antibiotic-mediated killing by altering the inflammasome-type I IFN axis. Our results also highlight this cross-regulation between two important innate response pathways with SRT boosting the macrophage pathogen control program by repressing a pro- pathogenic and activating the host beneficial response. In line with this observation, treatment with inflammasome inhibitors reversed the ability of SRT to enhance the effects of HR in macrophages. While our work points toward the type I IFN antagonism-mediated inflammasome activation as a key mechanism in providing synergy to anti-TB therapy, identifying the exact molecular target of SRT in this process remains an important future challenge.

SRT combination with front-line anti-TB drugs provided early bacterial control with improved resolution of pathology and enhanced host survival, thereby being a strong candidate for host directed therapy. SRT provides additional benefits as an adjunct modality. The pharmacological properties of SRT with excellent PK-PD, safety and tolerance for long term usage in the human population has been well established (*Sheehan and Kamijima, 2009*; *Ronfeld et al., 1997*; *Mandrioli et al., 2013*). Interestingly, two patients undergoing TB therapy with INH given SRT as an anti- depressant, did not show any deleterious effects on long-term use of the combination, auguring well for safety in the human population (*Malek-Ahmadi et al., 1996*; *Judd et al., 1994*). In addition, these studies combined with the enhanced protective capabilities of the combination therapy in pre-clinical animal models (our data), indirectly rule out any possibility of negative drug-drug interactions between SRT and ATT on prolonged usage. It is logical to expect that SRT with its wide use as an anti-depressant in adults and children may be beneficial in efficiently with a combination regimen of frontline TB drugs and SRT.

A recent clinical trial with SRT as an adjunct regimen with fluconazole for asymptomatic crypto-coccal antigenemia demonstrated SAEs (non-fatal) in 4 out of 10 participants (3 with psychosis and aggressive behavioral changes, 1 with serotonin syndrome) as opposed to 3 out of 10 (1 fatal) SAEs in the control placebo group (*Boulware et al., 2020*). While such concerns cannot be overlooked for use of SRT as an adjunct TB therapy, it would be prudent to weigh the advantages of the tackling the dual problem of severe drug induced depression in TB patients (invisible co-morbidity) (*Sweetland et al., 2017*; *Trenton and Currier, 2001*) in future detailed studies on the effective dose and duration of therapy needs to be undertaken. Our studies with lower doses of sertraline showing greater efficacy would be of immense use in future clinical studies for testing this adjunct regimen. However, the collective properties of a SRT adjunct TB therapy – faster bacterial control, enhanced host survival and capacity to target drug tolerant/ dormant bacterial populations augurs well for the highly constrained national/ global economy combating the TB pandemic.

## Materials and methods

Bacterial Strains and Growth Conditions—Mtb strains were cultured in Middlebrook 7H9 broth with 4% ADS or in 7H10/7H11 agar (BD Biosciences, USA) with 10% OADC (HiMedia laboratories, India).

### Reagents

THP1 Dual Monocytes was obtained from InvivoGen (Toulouse, France). The integrity of the cells were tested by STR profiling with routine mycoplasma testing. HiglutaXL RPMI-1640 and 10% Fetal Bovine Serum (HIMedia laboratories, Mumbai, India), PMA (Phorbol 12-Mysristate 13-acetate- P8139, Sigma Aldrich, USA), BX795 (tlrl-bx7, Invivogen) were used for culture of cells. The following reagents were procured from Sigma Aldrich, USA: Vit C (L- ascorbic acid, A5960), oleic acid albumin (O3008), Isoniazid (I3377), Pyrazinamide carboxamide (P7136), Ethambutol dihydrochloride (E4630) and Sertraline hydrochloride (S6319). Rifampicin (CMS1889, HIMEDIA laboratories, Mumbai, India) and commercially available SRT (Daxid, Pfizer Ltd, India) was used for mouse studies.

## Macrophage infection

THP1 Dual reporter monocytes were grown in HiglutaXL RPMI-1640 containing 10% FBS and differentiated to macrophages with 100 nM PMA for 24 hr. Following a period of rest for 48 hr, cells were infected with single cells suspensions (SCS) of Mtb at a MOI of 5 for 6 hr. For analyzing the Interferon (IRF pathway) activation levels, supernatants from Mtb infected THP1 Dual macrophages were assayed for stimulation by measuring luminescence as per manufacturer's recommendations. For IFNβ ELISA, an MOI of 10 was used for infection of THP1 cells continuously for a period of 24 hr. Cell supernatants were then used for ELISA as per recommended protocols (Human IFN-beta Duoset, R&D systems).

## Monocyte-derived macrophage culture

PBMCs were isolated from fresh blood obtained from healthy donors in accordance with Institutional human ethics committee approval (Ref no: CSIR-IGIB/IHEC/2017–18 Dt. 08.02.2018). Briefly, 15–20 ml blood was collected in EDTA containing tubes and layered onto HiSep (HIMedia laboratories, Mumbai, India) and used for isolation of PBMCs according to the recommended protocols. Post RBC lysis, cells were seeded at a density of $3x10^5$ cells/ well and differentiated into monocyte derived macrophages with 50 ng/ml GMCSF for 7 days and then used for infection with Mtb.

### Analysis of response parameters

For analysis of different parameters of cellular response to infection, qRTPCR based gene expression analysis and cytokine ELISA in culture supernatants were performed according to manufacturer's recommendations.

## Analysis of gene expression by qRTPCR

Total RNA was isolated from macrophages suspended in Trizol by using the recommended protocol. cDNA was prepared from 1 µg of RNA by using the Verso cDNA synthesis kit and was used at a concentration of 10 ng for expression analysis by using the DyNAmo Flash SYBR Green qPCR Kit (Thermo Fisher Scientific Inc, USA).

### Analysis of cytokine secretion by ELISA

Culture supernatants at different time intervals post infection/ treatment were filtered through a 0.2µ filter and subjected to ELISA by using the eBioscience (Thermo Fisher Scientific Inc USA) ELISA kit as per recommended protocols.

## Bacterial survival in macrophages

For determining intra cellular survival of Mtb strains macrophages were seeded in 48well plates and infected with Mtb at MOI 5 for 6 hr. SRT (20 µM), BX795 (10 µM) and TB drugs C1, C2, C3: (C1- INH-200ng/ml, Rifampicin-1000ng/ml, -C2 and C3:10- and 25-fold dilutions of C1) and used for treatment of macrophages at the appropriate concentrations. At specific days post infection macrophages were lysed with water containing 0.05% of tween80. Dilutions of the intracellular bacterial numbers were made in PBS with 0.05% of tween80 and plated on 7H10 agar plates. The VitC induced model of antibiotic tolerance in macrophages was developed as described earlier with cells treated with 2 mM Vit C for 24 hr and then with Isoniazid and rifampicin for a further 3 days (*Sikri et al., 2018*). For testing in lipid rich macrophages, cells were treated with oleic acid at 200 µM concentration after PMA differentiation for 2 days, infected with Mtb for 6 hr, followed by treatment for 5 days and enumeration of bacterial numbers.

## Mouse infection and antibiotic treatment

(6–10 weeks old) C3HeB/FeJ/ C57BL6/Balbc animals were infected with Mtb clinical isolate at 500 CFU per animal through aerosol route. Two/4 weeks post infection animals were started on antibiotics H (100 mg/l), R (40 mg/l) (*Vilchèze et al., 2018*), Z (150 mg/l), E (100 mg/l) (*Lanoix et al., 2016*) and SRT (10 mg/l, human equivalent dose of 3.3 mg/kg/day), as required treatment by giving all of the drugs ad libitum in their drinking water for 7 weeks which was changed twice every week. For survival, animals were monitored regularly and euthanized at a pre-determined end point according to the Institutional animal ethics approval. For estimating tissue bacterial burdens, lungs and spleen of infected animals

were collected in sterile saline, subjected to homogenization and used for serial dilution CFU plating on 7H11 agar plates containing OADC as supplement. Colonies were counted after incubation of the plates at 37 °C for 3–5 weeks and recorded as CFU/tissue.

All statistical analysis was performed by using student's t-test for significance, p values of<0.05 was considered significant.

## Acknowledgements

The authors thank CSIR (VR-BSC0123, OLP1136, MLP2106, MLP2012), and ICMR-ITRC (GAP0213) for supporting the study. CSIR-STS0016 is acknowledged for continuous maintenance of BSL3 and ABSL2 facilities. The student fellowships from CSIR, UGC and DBT India are acknowledged: DS- DBT JRF, AS- CSIR JRF, PA- CSIR-BSC0124, CSIR- SRF, SD-.DBT-JRF. The authors wish to thank Dr. Anurag Agrawal, CSIR- IGIB, New Delhi, India and Dr. Michael Glickman, MSKCC, New York, USA, for suggestions in improving the manuscript.

## Additional information

### Funding

| Funder | Grant reference number | Author |
| --- | --- | --- |
| Council of Scientific and Industrial Research, India | BSC0123 | Vivek Rao |
| Council of Scientific and Industrial Research, India | OLP0136 | Vivek Rao |
| Indian Council of Medical Research | GAP0213 | Vivek Rao |
| Council of Scientific and Industrial Research, India | STS0016 | Vivek Rao |
| Council of Scientific and Industrial Research, India | MLP2106 | Vivek Rao |
| Council of Scientific and Industrial Research, India | MLP2012 | Vivek Rao |

The funders had no role in study design, data collection and interpretation, or the decision to submit the work for publication.

### Author contributions

Deepthi Shankaran, Data curation, Formal analysis, Validation, Investigation, Methodology, Writing – original draft; Anjali Singh, Data curation, Formal analysis, Validation, Investigation, Methodology, Writing - review and editing; Stanzin Dawa, Supervision, Investigation, Methodology, Writing - review and editing; Prabhakar Arumugam, Validation, Investigation, Methodology, Writing - review and editing; Sheetal Gandotra, Conceptualization, Resources, Supervision, Validation, Methodology, Writing - review and editing; Vivek Rao, Conceptualization, Resources, Data curation, Formal analysis, Supervision, Funding acquisition, Validation, Investigation, Methodology, Writing – original draft, Project administration, Writing - review and editing

### Author ORCIDs

Sheetal Gandotra http://orcid.org/0000-0002-1204-7290
Vivek Rao http://orcid.org/0000-0001-8646-6634

### Ethics

Human subjects: The study was conducted in strict accordance with recommendations of the National Ethical Guidelines for Biomedical and Health Research Involving Human Participants, Indian Council of Medical Research (ICMR), Government of India. The protocols followed were approved by the Institutional human ethics committee of IGIB, proposal no 10, 2016 and Ref no: CSIR-IGIB/IHEC/2017-18 Dt. 08.02.2018. Patient informed consent was obtained prior to commencing the work.

This study was performed in strict accordance with the recommendations of the Control and Supervision of Experiments on Animals (CPCSEA). Experiments were conducted in line with approved protocols of the Institutional animal ethics committee, CSIR- IGIB (IGIB/IAEC/OCT2018/10, IGIB/IAEC/13/28/2020).

### Decision letter and Author response
Decision letter https://doi.org/10.7554/eLife.64834.sa1
Author response https://doi.org/10.7554/eLife.64834.sa2

---

## Additional files

### Supplementary files
• Transparent reporting form

### Data availability
This manuscript does not include any sequencing or high through put data. Also this work does not include any protein crystallization and diffraction analysis. The manuscript does not involve data for submission to depositing datasets into a domain-specific public archive.

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
