## [Editor Report]

Host-directed therapies have the potential to improve the management of tuberculosis by shortening the duration of chemotherapy and promoting recovery of respiratory sufficiency. In this useful study, the authors investigate the utility of sertraline as a potential host-directed therapy. They provide solid evidence that sertraline potentiates the activity of anti-tubercular drugs in macrophages as well as in the murine model of tuberculosis infection. The study will be of interest to tuberculosis researchers and infectious disease specialists.

---

## [Decision Letter]

**Decision letter after peer review:**

Thank you for submitting your article "The antidepressant sertraline provides a novel host directed therapy module for augmenting TB therapy" for consideration by *eLife*. Your article has been reviewed by 3 peer reviewers, and the evaluation has been overseen by Bavesh Kana as the Senior and Reviewing Editor. The following individuals involved in review of your submission have agreed to reveal their identity: Robert Wallis (Reviewer #2); Robert O Watson (Reviewer #3).

Essential Revisions:

Major concerns:

1. The conclusion that SRT enhances inflammasome activation is not supported by the data. The control Isoliquiritigenin, a NLRP3 inflammasome inhibitor, potentially has other effects. In addition, NLRP3 inflammasomes appear to enhance Mtb growth and spread in host cells (see Beckwith et al. 2020) which would counter the argument that NLRP3 inflammasomes are central to SRT mediated bactericidal effects. Hence studies to demonstrate that the activity of SRT can be directly linked to inhibition of signaling would corroborate the hypothesis that SRT acts as an HDT. In this regard, demonstrating that SRT acts by interfering with IRF3 signaling would be helpful in addition to demonstration that IRF3 transcription is downregulated. There are multiple tools and assays available to explore pathways that lead to IRF3-dependent transcription as well as the ultimate outcome of these effects on phagosome maturation or autophagy.

2. SRT is presented as an antagonist of polyI:C-induced type I IFNs. However, during TB infection, cytosolic DNA sensing via the cGAS/STING axis constitutes the major pathway through which type I IFNs are induced in macrophages. To offer more support that SRT inhibits type I IFN, the authors should consider measuring the actual amount of type I IFN using an IFNb ELISA. Additionally, the authors should use human/mouse primary macrophages (not just THP1 reporter cells) and measure transcript levels (at key time points post infection) and protein levels of type I IFN and other proinflammatory mediators (e.g. TNFa, IL-1, IL-6) +/- SRT to determine if SRT is specific to the type I IFN response. If this is indeed the case, other NFkB genes/cytokines should not be impacted. A set of experiments using an IFN blocking antibody would enhance Figure 2, as both cGAS and STING KO macs have significant differences in basal gene expression and their ability to respond to innate immune stimuli.

3. Figure 1. It is hard to understand what the relative contributions are of sertraline (SRT) compared to the other treatments from the way that the data is presented in panels C and D. It would be easier to look at the data if presented as in panel B. What is the concentration of SRT in panel B? from panel C, the effect of HR is worse on day 5 compared to day 3, which does not match with the expected efficacy of this two-drug combo: thus, at 0.2ug/mL Rif (10X MIC) and 1ug/mL Rif (50X MIC), the effect at day 5 should be better than day 3. The text mentions that the effect is greater on day 5 (92-fold reduction compared to 62 on d3) but this is not supported by the C1 bars in panel C. Similarly, in panel D, C1 is worse on da 5 compared to day 3, thus the statement that SRT improves potency for C1 doses is hard to interpret. Panel E: what is the treatment time for the MDMs? This seems to be missing from the Materials and methods as well. The HR treatment appears rather inefficacious for the MDM if this is the C1 concentration but maybe the treatment time is short.

4. Figure 1H: what macrophage model was used? Mtb CFU go up by 10-fold which is not comparable to any of the other panels. How long have the macrophages been treated? It is odd that there is no dose dependence between 1-5uM SRT despite remarkable potency at these concentrations. The SRT needs to be titrated to lower doses (eg., 0.5 and 0.1uM). Sertraline and BX795 both inhibit IRF3 dependent signaling but by different mechanisms. SRT inhibits IRF3 signaling through inhibition of the PI3K pathway. In Figure 2 panels B and C, there is little evidence that SRT and BX795 have similar effects from the way that the data is presented. It would be best to perform experiments with BX795 using a similar experimental design to that used for SRT and to present this as in Figure 1 (as requested above, using a layout as in panel B). It seems odd that BX795 by itself causes bacteriostasis but it does not cause bacteriostasis in combination with HR even though it potentiates the activity of HR (panel C).

5. Isoliquiritigenin is not a very specific immune modulator, having multiple effects including effects on NLP3 inflammasomes. Nevertheless, it has an effect on Mtb CFU in the macrophages. To strengthen this, experiments with SRT alone and SRT + isoliquiritigenin should be added.

6. Figure 2F should also include a treatment with HR. This will support a model where IFN-γ treatment augments SRT activity.

Other concerns:

in vitro studies (i.e. bacterial culture) were only performed with SRT up to 6 μm while the cultured cell experiments used a range up to 20 uM. 5 μm had almost no effect on the viability/growth of Mtb in macrophages. Why was such a low concentration used in the in vitro experiments and a very high concentration in cell culture. Related to this, will the dose that was used in mice, when translated to humans, lead to toxicity? The authors may wish to consider a recent study by David Boulware of adjunctive sertraline in patients with cryptococcal antigenemia that was stopped early due to psychoses and serotonin syndrome. Please include these sentiments in the discussion. Also, what is the physiological concentration of SRT when taken for depression and how does that compare to the concentrations used in vitro? Are the in vitro concentrations feasible to achieve in patients?

Vitamin C leads to growth arrest of Mtb in vitro whereas in macrophages, Vitamin C does not suppress growth but does render isoniazid, rifampicin and ethambutol ineffective (Sikri et al., 2018). Stating that these bacteria are dormant (lines 113 – 117) comes across as incorrect given that the Mtb cells are growing.

Figure 1G: 200uM oleic acid should cause most of the THP-1 cells to become necrotic. Presumably this is the case, but it's not clear what the status is of the cells since these are infected. Please clarify.

Why does HR have no effect on WT macrophages in Figure 2, panels D-E? The type of macrophage used in Figure 2 should be indicated since it is known that different macrophage lineages are quite different with respect to their type 1 IFN production.

Figure 3B: The secretion of TNF-α is hard to understand. It is dramatically increased only at 42h in the HRS sample but supressed compared to HR alone in the other time points.

In contrast, IL1beta secretion does not explain the activity of SRT. At 42h, it is a bit higher in the HRS sample compared to untreated, HR or SRT-only samples. However, at 66h, it is highest in the untreated sample. Can the authors explain this?

An infecting dose of 500 CFU, especially in C3HeB/FeJ mice, does not lead to formation of caseous granulomas, although this is strain-dependent. This is an incredibly high dose. What drove the selection of this dose? It seems that the authors set themselves up for less robust results by using such high doses. The strain of Mtb used is critical and never mentioned. In addition, the authors do not show whether the classical type 1, type 2 and type 3 lesions form, which would generally be the basis for the selection of this mouse model.

Delivering drugs via drinking water is not ideal but is sometimes necessary when there's no manpower to do daily oral gavage. SRT is given as 10ug/mL in the drinking water. Do the authors know what the serum drug concentration is? Does it affect the serum drug concentrations of the other anti-tubercular drugs? It is possible that SRT affects metabolism of the other drugs thus this information seems important to have.

Figure 4: please indicate the number of mice in each group.

SRT increases the survival of the HRC3 and HRC2 drug treatments groups in Figure 4. How does SRT affect organ burdens? Effects on lung pathology are shown in the HRC1 and HRC1+ SRT groups showing improved pathology. This group survived beyond day 90 but are the effects on bacterial organ loads known?

The statement "the adjunct regimen was efficient in controlling dissemination of infection into spleens" is not necessarily supported by the data. Mice are treated at week 4 by which time dissemination has already occurred. Thus, SRT is effective in both lungs and spleens as an adjunct treatment but it's not clear what effect it has on dissemination.

Although there is some discussion of the histological findings in Figures 5-6, the images do not really show much difference between the groups. The differences shown can merely be attributed to the differences in bacterial burdens and does not point to any host-directed effect. A key feature of potential HDTs would be to reduce lung injury. It was difficult to assess the effects of SRT on lung pathology. Perhaps a detailed Image analysis would be helpful.

The authors may wish to cite a study by Sarah Stanley in Plos Pathogens in 2014 that found intracellular anti-TB effects of fluoxetine.

It is unlikely that HDTs would be adopted for the purpose of reducing the dose of TB drugs, but more likely for shortening treatment. Do the authors have any data on relapse in mice?

It might be worth consistently using the more common INH and RIF abbreviations to increase the clarity/readability of the manuscript and figures.

In Figure 3B, why is there a spike in TNF-a in the HRS treated cells only at 42h?

Was statistical analysis performed on the data in Figure 3B and D?

A description/discussion of the different mouse strains use in infection – what benefits each has as a model and why several were used – would help convey the impact of the in vivo studies.

Since antibiotics and SRT were administered ad libitum, how did the authors ensure that mice took enough of the antibiotics and especially SRT? Is it known whether these drugs affect the water taste enough to affect a mouse's willingness to drink them?

Was statistical analysis performed on time-to-death experiments?

Were CFUs measured in mice from Figure 4 to determine empirically how effective the antibiotic treatments were? And if SRT impacted their effectiveness?

---

## [Author Response]

Essential Revisions:Major concerns:1. The conclusion that SRT enhances inflammasome activation is not supported by the data. The control Isoliquiritigenin, a NLRP3 inflammasome inhibitor, potentially has other effects. In addition, NLRP3 inflammasomes appear to enhance Mtb growth and spread in host cells (see Beckwith et al. 2020) which would counter the argument that NLRP3 inflammasomes are central to SRT mediated bactericidal effects. Hence studies to demonstrate that the activity of SRT can be directly linked to inhibition of signaling would corroborate the hypothesis that SRT acts as an HDT. In this regard, demonstrating that SRT acts by interfering with IRF3 signaling would be helpful in addition to demonstration that IRF3 transcription is downregulated. There are multiple tools and assays available to explore pathways that lead to IRF3-dependent transcription as well as the ultimate outcome of these effects on phagosome maturation or autophagy.

Isoliquiritigenin is recognised primarily as an inflammasome inhibitor (Honda et.al, , 2014, J Leukoc Biol. 96(6):1087-100), although its potential for other effects is also demonstrated. We agree with the reviewer that the previous study (Beckwith et.al., 2020) that inflammasome activation results is detrimental to bacterial growth, however, the importance of inflammasome in the context of TB-antibiotic treatment is not yet understood.

We employed two strategies to answer these questions:

1) To test if specific inflammasome activation was important for antibiotic efficacy, we used a more specific inhibitor of NLRP3 inflammasomes- MCC950 in HRS treated macrophages and observed reversal of antibiotic efficacy in these cells (Figure 3G).

2) By comparing antibiotic potencies in macrophages with and without MCC950 treatment (Fig-3H), we observed loss of antibiotic efficacy in MCC950 treated cells again supporting the synergy of inflammasome activation and antibiotics.

In parallel, we have also tested the effect of inflammasome activation by LPS and nigericin in improving the effectiveness of H and R in controlling bacterial growth in macrophages and observe that treating cells with LPS and nigericin augments antibiotic efficacy again substantiating the importance of inflammasome activation on antibiotic efficacy. We are currently pursuing this study forward and this would form the basis of a subsequent manuscript detailing the importance of this axis.

The IFN inhibitor BX795 is known to be a type I IFN inhibitor in cells (Clark et.al, 2009, J. Biol. Chem, 284(21): 14136-46). BX795 is recognised as a potent inhibitor of type I IFN signaling, selectively blocking the phosphorylation, nuclear translocation and transcriptional IRF3 signalling in RAW macrophages. BX795 was identified as a potent inhibitor of IRF3 nuclear translocation in a screen for small molecule modulators of STING- IRF3 (Koch et.al, 2018 ACS Chem Biol. 13 (4): 1044-81). We have employed BX795 in our study and show comparable effects of this treatment to SRT treatment on Mtb growth restrictions as well as antibiotic effectiveness providing evidence for type I IFN restriction via IRF3 signalling inhibition.

2. SRT is presented as an antagonist of polyI:C-induced type I IFNs. However, during TB infection, cytosolic DNA sensing via the cGAS/STING axis constitutes the major pathway through which type I IFNs are induced in macrophages. To offer more support that SRT inhibits type I IFN, the authors should consider measuring the actual amount of type I IFN using an IFNb ELISA.

We thank the reviewer for this suggestion. We have measured type IFN in macrophages and find that treatment with SRT significantly reduces type I IFN levels in macrophages to almost uninfected levels as shown in Figure 1B.

Additionally, the authors should use human/mouse primary macrophages (not just THP1 reporter cells) and measure transcript levels (at key time points post infection) and protein levels of type I IFN and other proinflammatory mediators (e.g. TNFa, IL-1, IL-6) +/- SRT to determine if SRT is specific to the type I IFN response. If this is indeed the case, other NFkB genes/cytokines should not be impacted.

We have tested the effect of SRT on the expression of these genes in human MDMs. We do not see any significant change in the transcript levels of TNF in these macrophages following treatment with SRT, however given the variables associated with human primary cells, the numbers that we tested were small and did not reach any significance (Author response image 1). Even in the mouse Bmdm from C57BL6 mice, while the trend of increased expression of IL1β and decreased IFN with SRT treatment was observed, these results did not reach statistical significance. However, in these cells, we observed increased expression of TNF and IL6 with SRT treatment indicating the upregulation of NFkb dependent pathways in these cells. For another inflammatory cytokine like IL18, again we did not see any significant difference in the levels of expression in the SRT- and SRT+ groups (Author response image 2). Given, our results with BX795, a potent inhibitor of IRF3 signalling and not impacting canonical NfkB signalling (Clark et.al, JBC, 2009), we shifted our attention to exploring the IFN 1 pathway in detail. Exploring detailed mechanistics of innate signalling affected by SRT would be part of a future comprehensive manuscript.

**Author response image 1. sa2fig1:** Relative expression of inflammatory cytokines in hMDMs treated with HR or HRS. Data depict fold expression (transcript abundance) relative to UI alone as mean + SEM from three independent experiments with triplicate wells each at 18h post treatment.

**Author response image 2. sa2fig2:** Relative expression of inflammatory cytokines in mBMDMs treated with HR or HRS. Data depict fold expression (transcript abundance) relative to UI alone as mean + SEM from two independent experiments with triplicate wells each at 18h post treatment.

In terms of cytokine secretion, we did observe a statistical downregulation of TNF and IFNα levels in human MDM, however, we did not observe any difference in the levels of TNF, IL6 or IFNγ in these cells (Author response image 3). Moreover, while we did not observe any significant difference in the levels of IL1β cytokine, the levels of IL1RA were significantly downregulated. Recent evidence suggests that IL1RA is involved in the stunting of IL1β signalling in line with our data of increased IL1β signalling and expression in the SRT treated cells. (JI et.al, 2019, nat. Microbiol. 4 (12): 2128-35).

**Author response image 3. sa2fig3:** Cytokine levels were determined with a Luminex-based multiplex assay. Average values + SEM at 18h and 42h post-treatment of three or two independent experiments of triplicate wells (N=2,3).

These results indicate that alteration of inflammatory response by SRT treatment is a dynamic process and would be cell dependent. While in the THP1 cells, these results are pronounced at the tested conditions, in primary human and mouse cells, the extent and kinetics of modulation may differ. Analysing these aspects in detail would involve significantly higher numbers of experiments that would form the basis for independent studies. While in most cases, a direct correlation of responses seen in one organism is possible, the inherent differences in the immune response dynamics the kinetics of this response can differentiate the responses seen in the two conditions. Moreover, given that the immune signalling pathways often intersect at multiple time points, it is often not possible to correctly associate a phenotype observed in one cell type vs the other.

A set of experiments using an IFN blocking antibody would enhance Figure 2, as both cGAS and STING KO macs have significant differences in basal gene expression and their ability to respond to innate immune stimuli.

We tested the effect of IFNAR antibody in an attempt to block the signalling cascade. While we did not see any previous reference to similar inhibition in any cellular model of infection, we also could not achieve inhibition of this pathway by this antibody. In response to infection and poly IC treatment (Author response image 4). While both cGAS and STING ko differ in basal gene expression, they play central roles in mediating cytosolic sensing in response to Mtb infections. Our results also suggest the importance of modulation of this pathway in SRT mediating augmented bacterial growth control. Detailed analysis of various signalling pathways would be the basis for a future manuscript to decipher the exact mechanism of this phenomenon.

**Author response image 4. sa2fig4:** IRF dependent luciferase activity in THP1 Dual. Cells were treated with IFNAR antibody for 2h following infection with mycobactreia and PIC treatment. Data depict luminescence relative to UI and UI + IFNAR Ab as mean + SD experiments with triplicate wells each at 3. 6, 24, 48 hr post treatment.

3. Figure 1. It is hard to understand what the relative contributions are of sertraline (SRT) compared to the other treatments from the way that the data is presented in panels C and D. It would be easier to look at the data if presented as in panel B. What is the concentration of SRT in panel B? from panel C, the effect of HR is worse on day 5 compared to day 3, which does not match with the expected efficacy of this two-drug combo: thus, at 0.2ug/mL Rif (10X MIC) and 1ug/mL Rif (50X MIC), the effect at day 5 should be better than day 3. The text mentions that the effect is greater on day 5 (92-fold reduction compared to 62 on d3) but this is not supported by the C1 bars in panel C. Similarly, in panel D, C1 is worse on da 5 compared to day 3, thus the statement that SRT improves potency for C1 doses is hard to interpret. Panel E: what is the treatment time for the MDMs? This seems to be missing from the Materials and methods as well. The HR treatment appears rather inefficacious for the MDM if this is the C1 concentration but maybe the treatment time is short.

We have modified the panels (now D and E) to make it easier to comprehend the contribution of SRT with respect to HR. SRT was used at the concentration of 20μM for macrophage infections, either alone or in combination with HR. We thank the reviewer for suggesting changes in depicting the figures, we have represented the data to read better. We do see a gradual increase in the efficiency of HR by day 5 than at day 3, however, while SRT in combination works 4-5 folds better than HR alone at day 3, the enhanced effect is not significant at day 5. We have used a concentration of 0.1X HR (HRC2) for MDMs again to evaluate if SRT would work in conditions of less-than-optimal efficacy of the frontline TB drugs. We have added this in the legends.

4. Figure 1H: what macrophage model was used? Mtb CFU go up by 10-fold which is not comparable to any of the other panels. How long have the macrophages been treated? It is odd that there is no dose dependence between 1-5uM SRT despite remarkable potency at these concentrations. The SRT needs to be titrated to lower doses (eg., 0.5 and 0.1uM). Sertraline and BX795 both inhibit IRF3 dependent signaling but by different mechanisms. SRT inhibits IRF3 signaling through inhibition of the PI3K pathway. In Figure 2 panels B and C, there is little evidence that SRT and BX795 have similar effects from the way that the data is presented. It would be best to perform experiments with BX795 using a similar experimental design to that used for SRT and to present this as in Figure 1 (as requested above, using a layout as in panel B). It seems odd that BX795 by itself causes bacteriostasis but it does not cause bacteriostasis in combination with HR even though it potentiates the activity of HR (panel C).

The THP1 infection model was used in this study of dose dependent efficacy of SRT (earlier Figure 1H, now Figure 2A). These are CFU values at day 5 in relation to day 0. This kinetics of growth is anticipated for Mtb in macrophages at an MOI of 5, wherein bacterial numbers increase until day 4-5 to values in the observed range. We again thank the reviewer for the suggestions to titrate the SRT doses to lower ranges. We have tested these at concentrations upto 0.05μM and have shown this as Figure 1A, Figure 2—figure supplement 1 (shown in Author response image 5, B).

Both BX795 and SRT inhibit type I IFN signalling although at different nodes in the signalling cascades. Experiments with BX795 at 10μM have been performed similar to the ones with SRT alone in THP1 macrophages. These results also show that similar to SRT alone, BX795 imparts bacteriostasis, while in combination with HR shows increased control than HR alone. While we are not able to completely comprehend the basis for this effect of both SRT and BX795, we feel that inhibition of Mtb driven type IFN induction after infection with Mtb, leads to an arrest of bacterial growth in macrophages. A detailed analysis of the underlying mechanisms responsible for this augmentation would shed further light into the actual pathway and would form the basis of a new manuscript.

**Author response image 5. sa2fig5:** Testing the effect of SRT at lower concentrations: A-IRF dependent luciferase activity in THP1 Dual macrophages following infection with Mtb at a MOI of 5. Cells were left untreated or treated with increasing concentrations of SRT for 24h in culture and the luminescence in culture supernatants was measured and is represented as mean + SEM from 2 independent experiments with triplicate wells each. (B) Growth of Mtb in macrophages infected with Mtb for 6h and then left untreated or treated with HRS at different concentration for 3 days. Data is represented as mean + SEM form 2 independent experiments containing triplicate wells per assay.

5. Isoliquiritigenin is not a very specific immune modulator, having multiple effects including effects on NLP3 inflammasomes. Nevertheless, it has an effect on Mtb CFU in the macrophages. To strengthen this, experiments with SRT alone and SRT + isoliquiritigenin should be added.

Again, we agree with the reviewer and have included in the manuscript (Figure 3F, inset). We also have included another NLRP3 specific inhibitor – MCC950 in the study (Figure 3G, H) that further substantiates the importance of host cell inflammasome in HR mediated control of infection in macrophages.

6. Figure 2F should also include a treatment with HR. This will support a model where IFN-γ treatment augments SRT activity.

This has been included in the manuscript as a supplementary – Figure 3- supplementary figure 1..

Other concerns:in vitro studies (i.e. bacterial culture) were only performed with SRT up to 6 μm while the cultured cell experiments used a range up to 20 uM. 5 μm had almost no effect on the viability/growth of Mtb in macrophages. Why was such a low concentration used in the in vitro experiments and a very high concentration in cell culture.

We haven’t seen any appreciable decrease in the growth of Mtb at upto 20μM in in vitro experiments, nearly 30-40% restriction after 8 days of culture. We used in combination of HR a lower dose of 6mM in combination with HR to offset the effect of minimal SRT inhibitory effects so that only the effect of SRT is understood.

Related to this, will the dose that was used in mice, when translated to humans, lead to toxicity? The authors may wish to consider a recent study by David Boulware of adjunctive sertraline in patients with cryptococcal antigenemia that was stopped early due to psychoses and serotonin syndrome. Please include these sentiments in the discussion. Also, what is the physiological concentration of SRT when taken for depression and how does that compare to the concentrations used in vitro? Are the in vitro concentrations feasible to achieve in patients?

We acknowledge the concerns of the reviewer and have incorporated these suggestions in the discussion about the possibility of excess toxicity and complications of using SRT at these doses. People with depression are prescribed between 50-200mg/kg/day, our dose ranges at the higher spectrum of the prescribed dose. To also address this further, we have done dose escalation experiments with lower doses of SRT in combination with TB drugs (Figure 6D, E).

Vitamin C leads to growth arrest of Mtb in vitro whereas in macrophages, Vitamin C does not suppress growth but does render isoniazid, rifampicin and ethambutol ineffective (Sikri et al., 2018). Stating that these bacteria are dormant (lines 113 – 117) comes across as incorrect given that the Mtb cells are growing.

The authors (Sikri et al., 2018) state that “Vit C-adapted bacteria display well-described features of dormancy, including growth stasis and progression to a viable but non-culturable (VBNC) state, loss of acid-fastness and reduction in length, dissipation of reductive stress through triglyceride (TAG) accumulation, protective response to oxidative stress, and tolerance to first line TB drug”. We have modified the text to reflect that the VitC model of THP1 infection reflects a antibiotic tolerant model of infection as suggested by the reviewer.

Figure 1G: 200uM oleic acid should cause most of the THP-1 cells to become necrotic. Presumably this is the case, but it's not clear what the status is of the cells since these are infected. Please clarify.

We have not seen any evidence of changes in the morphology of cells treated with 200μM Oleic acid for 2 days.

Why does HR have no effect on WT macrophages in Figure 2, panels D-E? The type of macrophage used in Figure 2 should be indicated since it is known that different macrophage lineages are quite different with respect to their type 1 IFN production.

HR was used at 0.1X concentrations. Hence the minimal bactericidal effect is observed. These are cGAS and STING mutant cell lines in the RAW264.7 mouse macrophage background. Previously, we have shown that Mtb induced type I IFN in these lines are comparable to THP1 cells (Shankaran et.al, 2022, J. Immunol.209(9):1736-1745). These details have been incorporated in the legends.

Figure 3B: The secretion of TNF-α is hard to understand. It is dramatically increased only at 42h in the HRS sample but supressed compared to HR alone in the other time points.In contrast, IL1beta secretion does not explain the activity of SRT. At 42h, it is a bit higher in the HRS sample compared to untreated, HR or SRT-only samples. However, at 66h, it is highest in the untreated sample. Can the authors explain this?

The authors wish to thank the reviewer for this query. We have reanalysed the data and have depicted the modified figures in the current text version. The spike at 42H for TNF was an oversight and due to an erroneous representation of the values in the figure. The new figure is shown in Author response image 6.

**Author response image 6. sa2fig6:** 

An infecting dose of 500 CFU, especially in C3HeB/FeJ mice, does not lead to formation of caseous granulomas, although this is strain-dependent. This is an incredibly high dose. What drove the selection of this dose? It seems that the authors set themselves up for less robust results by using such high doses. The strain of Mtb used is critical and never mentioned. In addition, the authors do not show whether the classical type 1, type 2 and type 3 lesions form, which would generally be the basis for the selection of this mouse model.

The authors agree that 500 CFU is a high dose of infection. However, our previous study has demonstrated that with lower doses (100 CFU), there is a high level of inherent variability amongst animals with respect to infection and the pathogenesis (Dawa et.al, Front Immunol, 2021 Sep 15;12:722735). Since our primary goal was to test the efficacy of SRT addition on a chronic infection model and at the level of host survivability, we used this dose that was consistent amongst individual mice in terms of infection and immunopathology. For a longer-term infection model in the more resistant mice-C57BL6/ Balbc, this dose did not affect the outcome of the assays.

We have used Mtb Erdman for routine drug sensitive and N73 for the drug tolerant studies.

Delivering drugs via drinking water is not ideal but is sometimes necessary when there's no manpower to do daily oral gavage. SRT is given as 10ug/mL in the drinking water. Do the authors know what the serum drug concentration is? Does it affect the serum drug concentrations of the other anti-tubercular drugs? It is possible that SRT affects metabolism of the other drugs thus this information seems important to have.

We preferred the use of ad libitum delivery of TB drugs in drinking water as used in the previous studies by Vilchèze et.al, 2018 Antimicrob Agents Chemother 23;62(3):e02165-17. Studies to check the PK PD, ADME etc and adverse effects of SRT on TB drugs are no being planned and will be the basis of a future study. A preliminary check of the potential interactions of SRT with frontline TB drugs has not been indicated (https://www.drugs.com/drug-interactions/sertraline.html). Moreover, two independent case studies with sertraline and INH have not shown any adverse effects. Future studies would help understand these in detail.

Figure 4: please indicate the number of mice in each group.

These have been indicated as suggested.

SRT increases the survival of the HRC3 and HRC2 drug treatments groups in Figure 4. How does SRT affect organ burdens? Effects on lung pathology are shown in the HRC1 and HRC1+ SRT groups showing improved pathology. This group survived beyond day 90 but are the effects on bacterial organ loads known?

We have not tested the effect of SRT on bacterial burdens on bacteria treated with HR alone as these studies were aimed at deciphering chronic pathology. We have tested the effect on bacterial loads in the C3HEBFEJ model with the four-drug therapy and the C57BL6 and Balbc models of infection.

The statement "the adjunct regimen was efficient in controlling dissemination of infection into spleens" is not necessarily supported by the data. Mice are treated at week 4 by which time dissemination has already occurred. Thus, SRT is effective in both lungs and spleens as an adjunct treatment but it's not clear what effect it has on dissemination.

We have modified the text as suggested.

Although there is some discussion of the histological findings in Figures 5-6, the images do not really show much difference between the groups. The differences shown can merely be attributed to the differences in bacterial burdens and does not point to any host-directed effect. A key feature of potential HDTs would be to reduce lung injury. It was difficult to assess the effects of SRT on lung pathology. Perhaps a detailed Image analysis would be helpful.

We again thank the reviewer for this insight. We have quantified the lesions in the lungs and have included it as a supplementary Figure 4—figure supplement 1.

The authors may wish to cite a study by Sarah Stanley in Plos Pathogens in 2014 that found intracellular anti-TB effects of fluoxetine.It is unlikely that HDTs would be adopted for the purpose of reducing the dose of TB drugs, but more likely for shortening treatment. Do the authors have any data on relapse in mice?

We have incorporated the study in the text. We have not tested the efficacy of SRT on the relapse model in mice.

It might be worth consistently using the more common INH and RIF abbreviations to increase the clarity/readability of the manuscript and figures.

We have used the conventional clinical abbreviations used for INH and Rifampicin

In Figure 3B, why is there a spike in TNF-a in the HRS treated cells only at 42h?

The authors wish to thank the reviewer for this query. We have reanalysed the data and have depicted the modified figures in the current text version. The spike at 42H for TNF was an oversight and due to an erroneous representation of the values in the figure. The new figure is shown in Author response image 6.

Was statistical analysis performed on the data in Figure 3B and D?

Yes, we have incorporated this information in the modified figure.

A description/discussion of the different mouse strains use in infection – what benefits each has as a model and why several were used – would help convey the impact of the in vivo studies.

These have been incorporated in the text.

A discussion of the mouse strains and their immunopathology in infection has been included in the text.

Since antibiotics and SRT were administered ad libitum, how did the authors ensure that mice took enough of the antibiotics and especially SRT? Is it known whether these drugs affect the water taste enough to affect a mouse's willingness to drink them?

We preferred the use of ad libitum delivery of TB drugs in drinking water as used in the previous studies by Vilchèze et.al, 2018 Antimicrob Agents Chemother 23;62(3):e02165-17. To avoid non drinking, we used 5% glucose in the water of all animals including the non-antibiotic treated groups. We also followed the uptake of water during the treatment and found comparable levels of usage between the groups.

Was statistical analysis performed on time-to-death experiments?Were CFUs measured in mice from Figure 4 to determine empirically how effective the antibiotic treatments were? And if SRT impacted their effectiveness?